# Optimisation strategies for directed evolution without sequencing

**Jessica James** [ID] [☾], **Sebastian Towers** [☾], **Jakob Foerster**, **Harrison Steel** [ID] *

Department of Engineering Science, University of Oxford, Oxford, United Kingdom

☾ These authors contributed equally to this work.
* harrison.steel@eng.ox.ac.uk

## Abstract

Directed evolution can enable engineering of biological systems with minimal knowledge of their underlying sequence-to-function relationships. A typical directed evolution process consists of iterative rounds of mutagenesis and selection that are designed to steer changes in a biological system (e.g. a protein) towards some functional goal. Much work has been done, particularly leveraging advancements in machine learning, to optimise the process of directed evolution. Many of these methods, however, require DNA sequencing and synthesis, making them resource-intensive and incompatible with developments in targeted *in vivo* mutagenesis. Operating within the experimental constraints of established sorting-based directed evolution techniques (e.g. Fluorescence-Activated Cell Sorting, FACS), we explore approaches for optimisation of directed evolution that could in future be implemented without sequencing information. We then expand our methods to the context of emerging experimental techniques in directed evolution, which allow for single-cell selection based on fitness objectives defined from any combination of measurable traits. Finally, we explore these alternative strategies on the GB1 and TrpB empirical landscapes, demonstrating that they could lead to up to 19-fold and 7-fold increases respectively in the probability of attaining the global fitness peak.

## Author summary

The standard approach to sorting-based selection in directed evolution is to take forward only the top-performing variants from each generation of a single population. There are, however, many possible approaches to exploring non-convex evolutionary fitness landscapes, and choosing this strategy as default may not always be the strongest approach. In this work, we begin to explore alternative selection strategies within a simulated directed evolution framework. We propose "selection functions", which allow us to tune the balance of exploration and exploitation of a fitness landscape, and we demonstrate that splitting a population into sub-populations can improve both the likelihood and magnitude of a successful outcome. We also propose strategies to leverage emerging selection methods that can implement single-cell selection based on any combination of measurable traits. We finally assess the space of alternative directed evolution strategies on the empirical

**Data Availability Statement:** Data and code used for running experiments and plotting is available on Software Heritage Archive at https://archive.softwareheritage.org/swh:1:dir:d61e1c81185705659c6ed545a77e7c84cd4aa6d5,

as well as GitHub at https://github.com/nesou2/direvo_sim.git. Raw data available at https://doi.org/10.5281/zenodo.12799997.

**Funding:** H.S. is supported in part by the Engineering and Physical Sciences Research Council (EPSRC, https://www.ukri.org/councils/epsrc/) projects EP/W000326/1 and EP/X017982/1. The funders did not play any role in the study design, data collection and analysis, decision to publish, or preparation of the manuscript.

fitness landscapes of the GB1 immunoglobulin protein and of TrpB tryptophan synthase. Our resulting proposal is that researchers should consider moving away from the standard approach, which we find to be generally sub-optimal, and implement population splitting to improve experiments. With improved knowledge from fitness landscape inference, directed evolution strategies could be further tailored using the tools proposed here.

## Introduction

Engineered biological systems hold immense potential for application across industries including medicine, manufacturing, and agriculture [1–3]. In recent decades, protein engineering in particular has demonstrated the potential of natural biological elements to be adapted for new functionalities. Advancements in computational methods are bringing *de novo* protein engineering closer to reality [4–8], however such approaches remain limited by our developing understanding of protein sequence-to-function relationships. To circumvent the need for such detailed prior knowledge, a range of techniques termed "directed evolution" have been developed [9]. Directed evolution techniques have delivered products across a range of applications, from cancer and autoimmune disorder drugs [10] to enzymes for converting cooking oil into bio diesel [11].

In a process that mimics nature, directed evolution consists of iteratively introducing random variation by mutagenesis, followed by selection biased toward desirable user-defined traits. Directed evolution methods are highly varied, and exist for both *in vivo* and *in vitro* settings. In this work, we have modelled *in vivo* directed evolution, in which fitness is evaluated inside a host organism such as a bacterial cell. Selection of living cells can be achieved broadly via two families of approaches; first, those that couple a trait-of-interest to growth of a host organism [12, 13], or second, those that couple a trait to a measurable output (e.g. expression of fluorescent proteins). In the latter, cells are screened to identify and isolate those with desirable output expression levels for future mutagenesis and propagation [14]. Each approach has different strengths and weaknesses. Growth-coupled selection utilises (comparatively) straightforward growth-based assays but requires engineering of trait-to-growth coupling, which is both challenging and can lead to "cheating" behaviour [15]. Meanwhile, screening-based methods only require a trait be *measurable* (i.e. they do not require coupling to growth), but in turn necessitate more complex experimental approaches to implement the selection for the measured winners (e.g. FACS [14]). In contrast to FACS, which takes only a single time-point measurement from each cell, emerging selection techniques leverage microfluidics to observe cells over long time periods prior to sorting [16, 17]. This produces large amounts of additional temporal information on which to sort cells, and allows selection to be performed based on dynamic phenotypes. Our work focuses on these emerging microfluidics-based methods due to their increased level of control, and consequently not all strategies we propose are suitable for growth-coupled directed evolution. This is a timely challenge, as new methods for single-cell selection pose novel theoretical questions—which our work aims to explore—regarding how their capabilities can be optimally implemented and exploited.

Mutagenesis methods are another essential component of directed evolution that accelerate the speed of evolution by introducing mutations at a greater than natural rate. *In vivo* mutagenesis methods introduce mutations to DNA inside living cells, which can then be cycled directly back into the selection process, resulting in a continuous and therefore less labour-intensive form of directed evolution. For example, methods including treatment with UV or chemical mutagens can lead to genome-wide increases in mutation rates, however, such

untargeted mutations may not always impact the trait of interest, and may cause unwanted effects elsewhere. An emerging family of more sophisticated targeted *in vivo* mutagenesis methods allows this limitation to be overcome in cases where specific sequence(s) are hypothesised to hold potential for improving a desired trait, by targeting mutagesis primarily within that region. Such methods, including EvolvR [18], MutaT7 [19], and OrthoRep [20], have already begun to be adopted by the wider community, and are continually undergoing further refinement for increased performance [21].

As a biological entity undergoes directed evolution, the process can be imagined as navigation across high-dimensional "fitness landscapes" [22]. Fitness landscapes map each genetic sequence to a measure of fitness (with "fitness" being performance for a desired function), and the goal of directed evolution is to find the highest peaks on that landscape. Fitness landscapes are known to exhibit variable degrees of ruggedness, which can create local optima that constrain paths of evolution [23–25]. Standard practice in directed evolution is to take forward and mutate only the top fraction of variants during each iteration [26, 28–31]. This "greedy" approach is prone to getting trapped in local optima, particularly on rugged landscapes [32]. With the aid of computational methods, however, it is possible to navigate protein fitness landscapes in a more active way. One of the earliest examples of such a method is ProSAR, which uses a statistical algorithm to identify specific residues that are correlated with high fitness. Each new generation of variants is designed to combine residues that were predicted to contribute most to fitness [33]. Methods that predict the fitness effects of mutations in this way are now able to accommodate machine learning [34–36] and Bayesian optimisation approaches [37, 38]. Such methods have been built upon by not only utilising the fitnesses of the sequences in isolation, but also time-series mutation data acquired *during* a directed evolution experiment [39]. The vast majority of these methods, however, are based upon the requirement for sequencing information from each generation. This means they are somewhat resource- and labour-intensive, and are not suited to maximising the benefits from *in vivo* mutagenesis methods for directed evolution [21].

In instances where sequencing information is unavailable, the field has established the default approach of selecting only the top variants with each generation [26, 28–31]. This approach may restrict population diversity, leading to a higher propensity to get trapped in local optima. Examples of previous work to address this include alternating between "on" and "off" states of selection [40], as well as modifying selection stringency [41] to increase population diversity. Here, we approach the challenge from several new angles. First, probability of selection is applied as a parameterised function of fitness that can be used to tune the balance exploration and exploitation on a fitness landscape. Second, we investigate the benefits that can be gained by splitting a population into sub-populations and allowing their trajectories to diverge. Finally, we explore the novel capabilities of the aforementioned emerging selection methods [16, 17], which enable effective optimisation of multiple properties in parallel. We assess the performance of our optimisation approaches by simulating directed evolution on the GB1 and TrpB empirical landscapes [23, 25].

## Results

In order to test selection strategies, a computational model was implemented to mimic the process of directed evolution. Genes in the model are represented by one-dimensional arrays, which iterate through rounds of mutation and selection (Fig 1, Methods: Model). In the selection process, the fitness of each gene is calculated using an empirical landscape [23, 25] or an NK model [42, 43]; see Methods. Empirical landscapes are combinatorially complete fitness measurements for all variants of a protein (or protein region). NK models are computationally

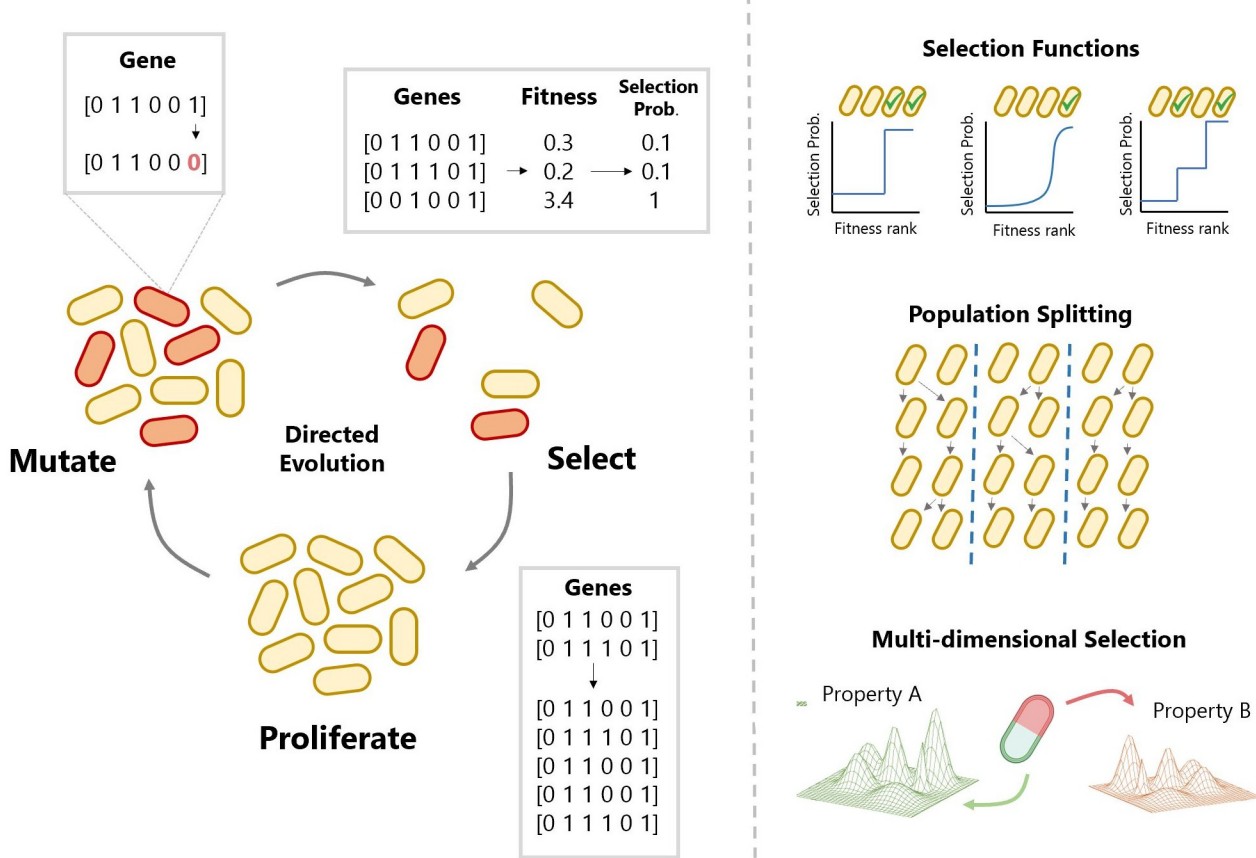

**Fig 1. Schematic of the directed evolution simulation cycle.** The model of directed evolution performs iterative round of mutation, selection and proliferation. Genes are represented by one-dimensional arrays. The fitness of each gene can be generated by feeding the array into a fitness landscape model. The probability of selection is determined by feeding the resulting fitness into a selection function. Proliferation is carried out by sampling with even probability up to a fixed population size. Mutation is carried out by introducing random changes to the arrays. For a more detailed description of the computational pipeline, see Methods: Model. Strategies explored using the model include selection functions, population splitting and selection across multiple properties.

generated fitness landscapes that are made necessary by the limited availability of empirical landscapes. In both NK and empirical landscape implementations, the gene sequence is taken as input and the corresponding fitness value is given as output. As outlined above, the approaches explored with the model are possible in contexts where one actively selects "winning" variants to enrich (via FACS or microfluidic sorting), as opposed to growth-coupled directed evolution, which does not offer this type of control.

## Selection functions for tuneable exploration vs exploitation

Selection functions are introduced as a means to tune the balance of exploration and exploitation on a fitness landscape. The selection functions proposed here are defined by two parameters: "fitness threshold" and "base chance" (Fig 2A). The fitness threshold is the fitness percentile above which variants have a 100% chance of selection, otherwise the chance of selection is equal to the base chance (Methods: Selection Functions). In this work, selection functions are normalised to select a constant fraction of variants. In a continuous directed evolution experiment, this ensures that proliferation time remains approximately constant

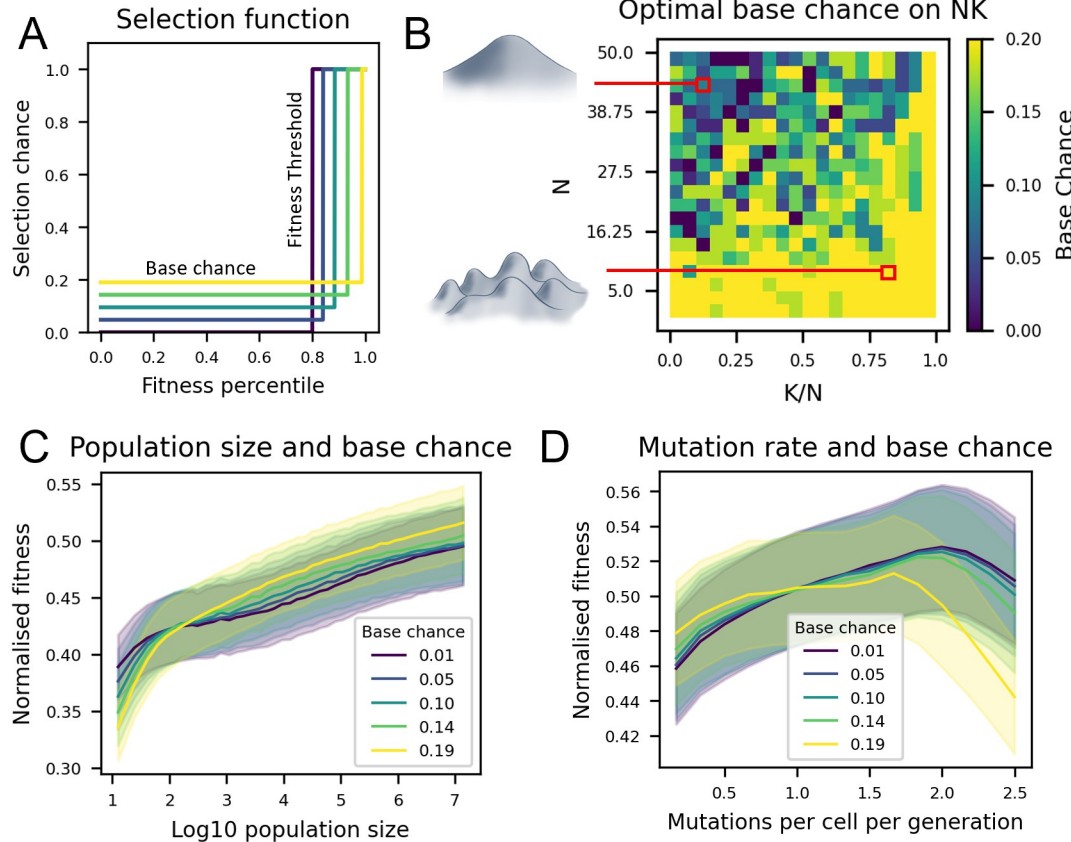

**Fig 2. Investigating the performance of selection functions in directed evolution.** A: Selection functions define probability of selection as a function of fitness (Methods: Selection Functions). The selection function used here is determined by two parameters: fitness threshold and base chance. Selection functions were normalised to select 20% of the population. B: Optimal base chance values on varying NK landscapes. Dependence of trajectory end-point fitness on C: population size, and D: mutation rate. "Normalised fitness" is maximum fitness across the population at the final time point (averaged over 100 runs) as a fraction of the global maximum on the landscape. Shaded areas represent standard deviation over repeat runs. Experiments ran for 300 generations. $0.01 \leq$ base chance $\leq 0.19$, N = 25, K = 5, mutations per cell = 0.1, population size = 1000.

between generations (hence, performance metrics can be considered as improvement in a trait *per unit time*). Fixing the selected proportion of cells also reduces the parameter space of selection functions to one dimension, as every base chance has only one fitness threshold value corresponding to a fixed proportion of the population being selected. We hypothesise that the base chance parameter will improve directed evolution by allowing a population to escape local optima on the fitness landscape. By selecting some cells unconditionally, they are allowed to accumulate more mutations, potentially allowing them reach higher performing variants via deleterious phenotypes.

The NK model describes a class of fitness landscape with tuneable ruggedness [42, 43] (Methods: NK Landscapes). N describes the number of variable sites, and K can be thought of as a metric of ruggedness ranging from 0 to $N - 1$, with high K meaning high ruggedness (i.e. more local optima). Fig 2B shows the optimal base chance over varying NK landscapes, as estimated via simulation. In particular, Fig 2B shows the optimal base chance increasing with respect to K (ruggedness), and decreasing with respect to N (dimensionality). Given that base chance is hypothesised to help escape local optima, one explanation for this result is that more

local optima are found in rugged landscapes, and they are more difficult to escape in low-dimensional landscapes, which offer fewer paths between any two points. For smooth, high-dimensional landscapes the opposite is true, therefore the function that most favours exploitation over exploration (i.e. base chance = 0) is found to be optimal. From this point onwards, the landscape N = 25, K = 5 was used as our standard landscape, as it displayed neither overly rugged nor overly smooth characteristics (S1 Fig). The robustness of base chance to noise on this standard landscape is displayed in S2 Fig.

Next, the interaction between base chance and population size was explored. Note that the NK landscape is non-linear, therefore a 1% increase in raw fitness may truly represent a larger underlying improvement, particularly at the high-fitness end of the distribution. Fig 2C shows that base chance can improve performance across a range of population sizes, particularly in large populations. Given that large populations can explore more of the landscape, they are likely to encounter more local optima. The ability for a population to escape a local optimum is dependent on its ability to reach a fitter state via at least one deleterious mutation. Evolution via deleterious mutations is shown to occur more readily in large populations [44, 45] and this dynamic may be promoted by base chance, hence the improvement in performance. In small populations, the cost of including detrimental variants is greater relative to the potential gain, therefore base chance is less beneficial.

The performance of the selection functions with varying mutation rates was also investigated (Fig 2D). When mutation rate is low, higher base chance values perform best and vice versa for high mutation rate. One explanation for this is in the balance of exploration and exploitation. Both base chance and mutation rate aid in escaping local optima by increasing the likelihood of a cell undergoing multiple mutations. Although this benefits exploration, *further* increases in mutation rate can come at the cost of not effectively exploiting a position on the landscape. For this reason, base chance performance drops off more quickly in a high mutation rate regime. Given that most directed evolution experiments operate in a low mutation rate regime to avoid detrimental side effects [46], base chance could act as a useful tool for promoting landscape exploration. The benefits of such an approach are that implementing a base chance has no direct impact on top-performing variants, whereas increasing mutation rate impacts all variants.

## Population splitting for improved exploration

Until now, this work has assumed a single population undergoing directed evolution. However, in practice, one could run multiple, smaller, copies of the same directed evolution experiment by subdividing a population, and take the best outcome across all of them as the final result. Here, this method is referred to as population splitting. An example of such a situation is displayed in Fig 3A, where a population of size 500 is split into five equal sub-populations of size 100. In this example experiment population splitting performs better, and Fig 3D and 3E demonstrate the consistency of this result across parameter regimes. This may be because in a single, mixed population, mutations will eventually drift to fixation or extinction, therefore the population as a whole remains largely on the same trajectory. If one splits the population, sub-populations are able to drift on separate trajectories without cross-competition, effectively mimicking the process of speciation and increasing landscape exploration [47, 48].

Principal components analysis (PCA) was used to verify that sub-populations diverge on separate trajectories. The final timepoint gene sequences from the simulation in Fig 3A were collected. PCA was performed on the combined dataset to reduce the 25-dimensional genetic space to just 2 dimensions for visualisation (Fig 3B). The result shows that each sub-population forms a cluster, and the overall variation of the split population is significantly more than the

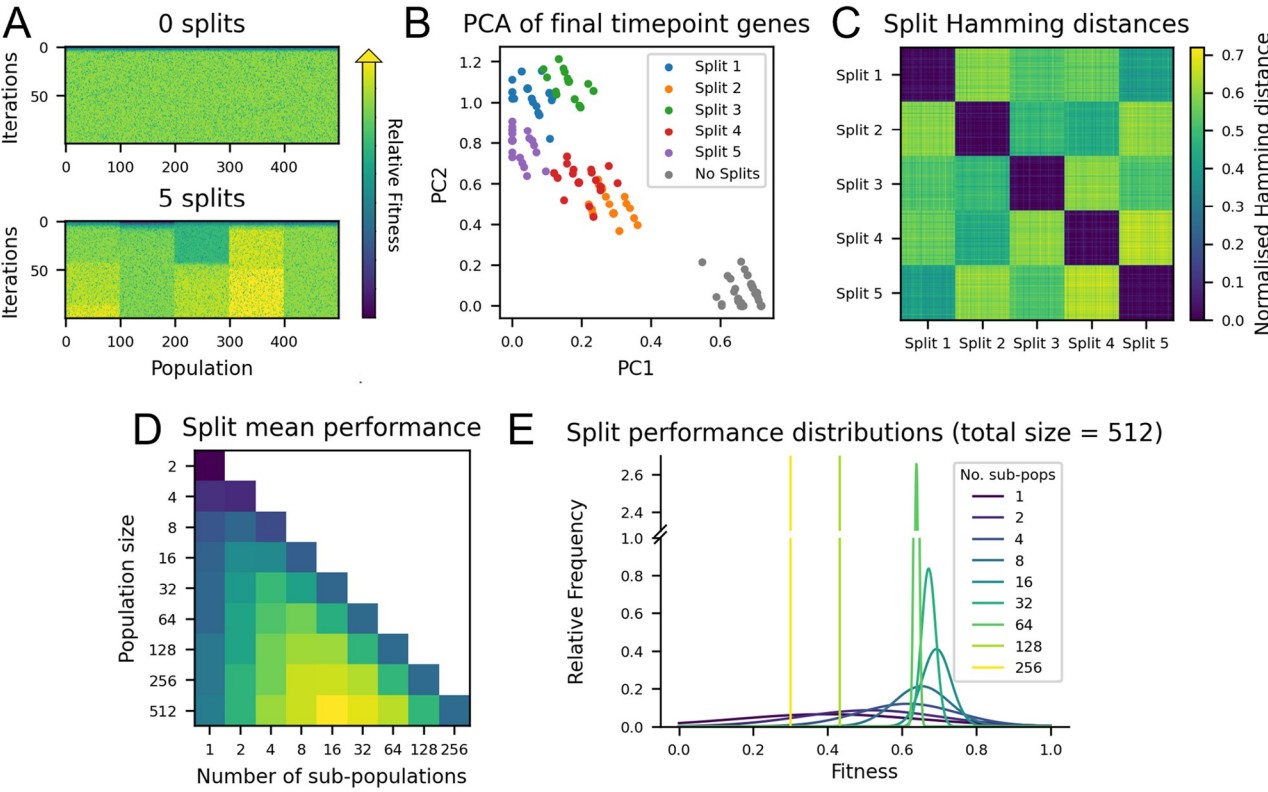

**Fig 3. Investigating the effects of population splitting in directed evolution.** A: Example of a directed evolution run split into five sub-populations vs a single large population. Colourbar indicates fitness value. B: Principal components analysis of the final time point sequences of a split vs non-split population. C: Hamming distances between the final timepoint sequences of the split population. D: Mean performance of directed evolution with varying total population and sub-population size. E: Mean and standard deviation of performance with varying number of sub-populations (fit to a normal distribution). Experiments ran for 100 generations. Mutations per cell = 0.1, base chance = 0, N = 25, K = 5.

non-split population. This is further verified in Fig 3C, which displays that the average normalised Hamming distance between sub-populations (0.44) was far greater than that within sub-populations (0.011, similar to the average Hamming distance of 0.009 measured within the large single population).

Population splitting can clearly confer a benefit to performance, however there is a trade-off between splitting the population to maximise exploration, and keeping sub-populations large enough to effectively search around their local position on the landscape. Fig 3D summarises the performance of population splitting for different total population sizes, demonstrating that if the (total) population is too small, splitting is instead detrimental to performance. An equivalent plot on landscapes of different ruggedness is displayed in S3 Fig. Fig 3E shows the distribution of performance outcomes, over 1000 runs, for splitting a large population into increasingly smaller populations. This demonstrates an additional advantage of population splitting, which is that the variance of the final outcome decreases as the number of sub-populations increases.

## Multi-dimensional selection with simulated novel selection methods

Previous sections have operated within the constraints of well-established directed evolution selection methods (e.g. FACS). Emerging methods for selection, however, may offer increased capabilities; notably using microfluidics to observe cells for long time periods prior to selection

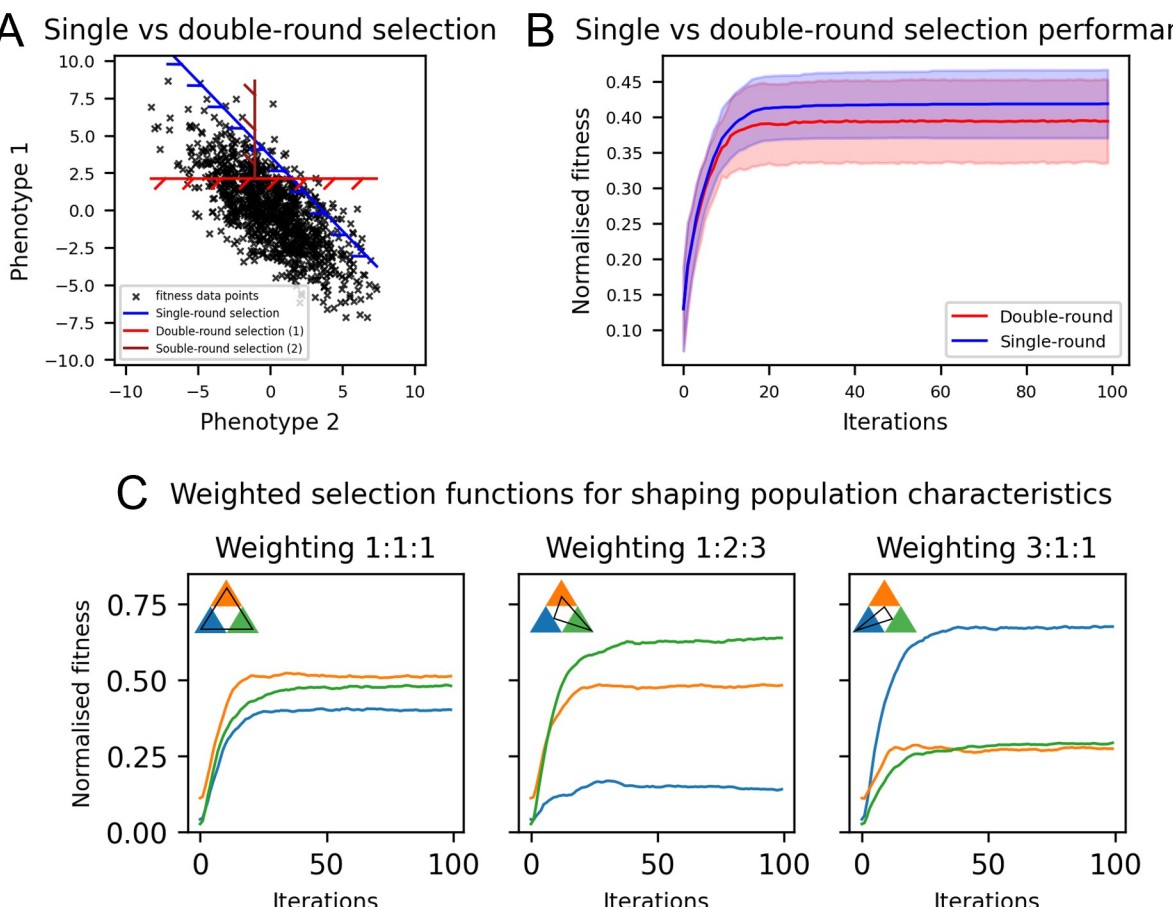

**Fig 4. Optimisation of multi-dimensional selection.** A: Demonstration of selection patterns with double-round (i.e. flow cytometry selection) and single-round selection. Randomly-generated fitness points normally-distributed around the mean 0 of phenotypes 1 and 2. B: Directed evolution performance of double-round vs single-round selection. Shaded region is standard deviation over repeated runs. C: Performance of weighted selection functions to perform directed evolution over three properties, with weightings of 1:1:1, 1:2:3 and 3:1:1. Experiments ran for 100 generations. N = 25, K = 5, mutations per cell = 0.1, population size = 1000, fitness threshold top 5%, as per [26].

[16, 17]. Not only would long-term observation increase the reliability of readings and allow selection based on complex time-dependent traits, it would also allow for multiple properties (or responses to stimuli) to be measured in a single round of selection. Such multi-dimensional selection is highly applicable in the directed evolution of biosensors, in which one seeks to optimise both specificity and sensitivity [26].

The current standard approach to multi-dimensional selection (e.g. in FACS) is to perform sequential rounds of selection, one for each property [26, 27]. The sub-optimality of this approach is previously highlighted in [53]. This introduces a systematic error with respect to the selection objective, as shown in Fig 4A. The blue dashed line divides the true best cells from the population (as could be achieved by single-round selection), whereas the red dashed lines represent the cut-offs of a double-round selection setup. Cells that are poor in one property but excel in another are not selected by double-round selection. As a result, the overall performance of single-round selection is higher (Fig 4B).

Not only would the described emerging methods improve upon FACS in the simplest case, they would also offer the additional ability to *tune* the prioritisation of different properties. When selecting on a single property, the fitness value (*F*) used to determine selection is simply

the value of that property. When selecting on multiple properties, however, the overall fitness value used to determine selection is some combination. Given that most directed evolution experiments have a limited amount of time and/or physical resources, it is crucial to consider how much one prioritises each property in this combination. This prioritisation can be implemented by applying a weight ($w_i$) to the value of each property. So, $F = w_1 f_1 + w_2 f_2 + w_3 f_3$, where $f_i$ is the value of property $i$. In the simplest case, we allow all weightings to be equal (Fig 4C). By changing the weightings of the properties we observe proportional gains in the fitness of each property.

## Translation to empirical fitness landscapes

The preceding results are based on simulations of NK landscapes, which although widely used, may not capture all important properties of real fitness landscapes. In order to test these strategies further, we therefore applied them two different empirical landscapes. Firstly the GB1 fitness landscape [23], which measures the binding strength of 149,631 variants of the GB1 immunoglobulin protein to IgG-Fc. Second, the TrpB empirical landscape [25] which measures the activity of 159,129 variants of typtophan synthase. In both GB1 and TrpB, a model was generated to impute the fitness of missing variants, bringing both combinatorially complete landscapes up to 160,000 variants (corresponding to 20 amino acid possibilities at each of the 4 sites).

We used these landscapes to assess the performance of strategies that employ base chance and/or population splitting. Fig 5A displays the performance of varying combinations of base chance and splitting. We observed a clear optimum in population splitting in the range of 50 sub-populations, which remains the optimum at most base chance values. The trends with respect to base chance are much weaker, and also dependent on the number of sub-populations. The impact of base chance is stronger when there is no population splitting.

Fig 5B displays the distribution of outcomes from GB1 directed evolution using four different strategies highlighted in Fig 5A (neither splitting nor base chance, each strategy in isolation, and both strategies combined). Of these four methods, the standard approach to directed evolution (employing neither splitting nor base chance) performs the worst, with over 60% of directed evolution runs on GB1 getting trapped at a local optimum (5.8x wildtype fitness). By introducing base chance that fraction is reduced to less than 30%, and by introducing splitting it is reduced to almost 0%. As for the fraction of runs which reach the *global optimum* (9.9x wildtype fitness), this reached 19% with the combined splitting and base chance approach compared to 1% using the standard approach, a 19-fold improvement.

Fig 5C and 5D display the equivalent measurements for the TrpB landscape. Compared to a strategy that employed neither base chance nor splitting, the optimal combined strategy generated a 7-fold increase in the probability of reaching the global optimum. Interestingly, greater benefits were achieved by base chance alone than by splitting alone, in contrast to GB1. The increased role of base chance in TrpB directed evolution could be on account of the fact that it is estimated to have 5x more local optima [25]. This may suggest that the role of base chance is more important in escaping local optima, whereas splitting strategies are more beneficial in terms of global exploration.

Despite the fact that GB1 and TrpB come from biologically very different contexts, they both responded best to a strategy that employs a combination of base chance and population splitting. Conversely, both exhibited the weakest performance with a strategy that employed neither approach (i.e. a typical "greedy" approach to directed evolution). Considering these observations, we hypothesise that the standard approach of no splitting and no base chance may *in general* significantly under-perform in real-world experiments. Deviating from the

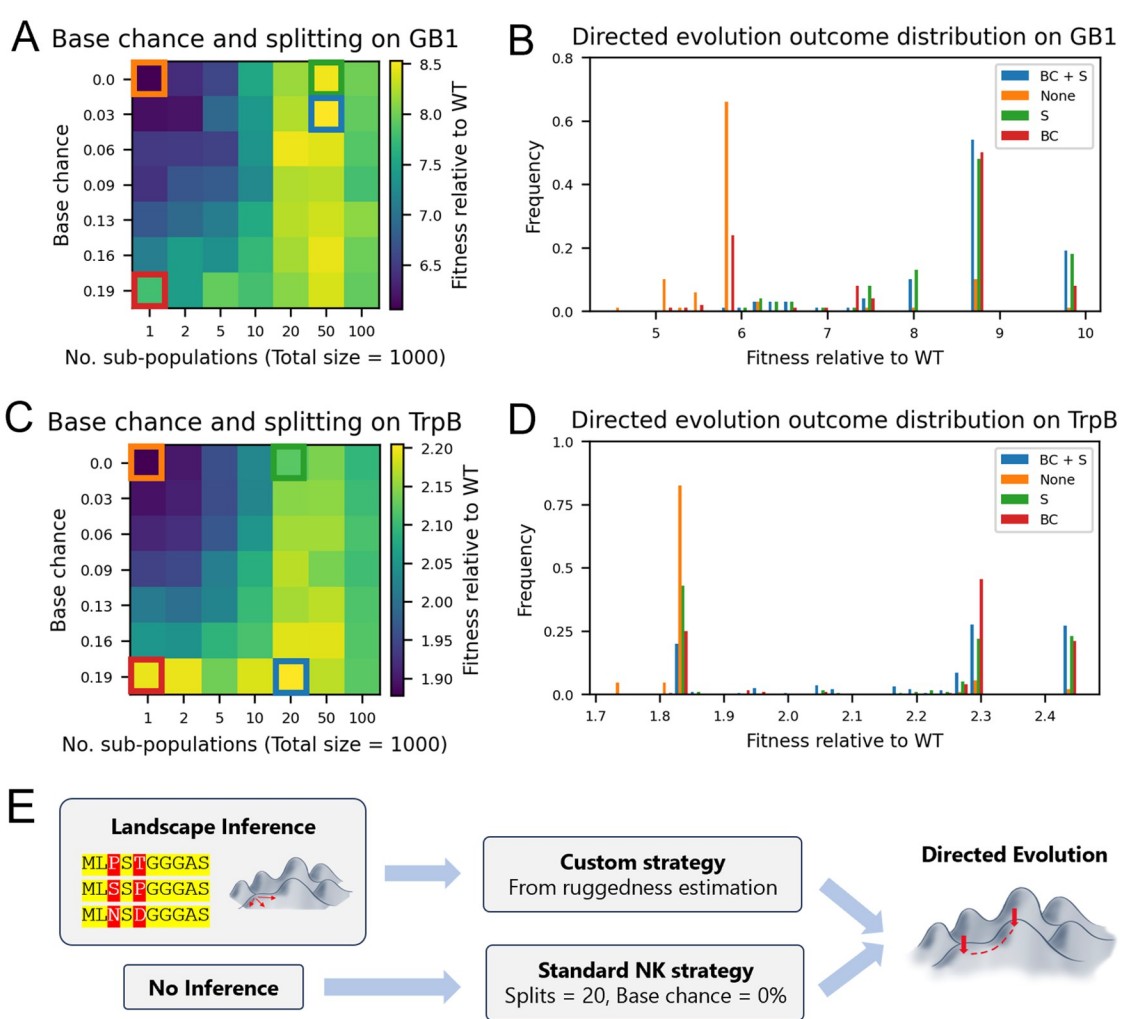

**Fig 5. Application of base chance and population splitting to the empirical GB1 and TrpB landscapes.** Mean performance (A) and outcome distribution (B) of varying base chance values and sub-population numbers in GB1 directed evolution. (C), (D): Same data respectively for TrpB. BC: base chance, S: splitting. Fitness score relative to WT (GB1: VDGV, TrpB: VFVS). Outcomes are from 1000 simulations with mutation rate = 0.01, total population size = 1000. E: Proposal directed evolution pipeline. Standard NK strategy comes from parameter sweep in S4 Fig.

standard approach to directed evolution, in particular by population splitting and/or adding a non-zero base chance, may therefore offer benefits even if it is not the absolute optimal strategy.

In order to determine what an improved default approach might look like, a parameter sweep over many different NK landscapes, population sizes and mutation rates was conducted to identify the most robust strategy (S4 Fig). A range of landscapes from completely smooth to rugged (K/N = 0.25) was selected. In the absence of a more informed prior, we weighted each landscape equally to calculate a final mean score. This embeds an assumption about the properties of real landscapes that might be considered in such experiments, which might be refined in future work as more experimental datasets become available. Following this simulation, our results suggest that in the absence of any prior knowledge, employing splitting but no base chance (up to 20 splits, 0% base chance) may improve likelihood of a successful outcome. On average, this strategy was in the 65th percentile of all strategies, compared to the 42nd

percentile for the standard no splitting approach. This "optimal" splitting level is comparable to that observed for GB1 and TrpB. However, base chance was found to be generally disadvantageous even at low splitting (contrary to observations on GB1 and TrpB) which may be due to the chosen distribution of landscapes simulated (which includes completely smooth landscapes). The fact that base chance appears beneficial on the empirical, non-smooth landscapes of Fig 5 also supports this suggestion. Nonetheless, attempting to implement 0% base chance accounts for the possibility of a highly additive landscape, and in practise, a certain degree may arise naturally due to inaccuracies in the selection process. In summary, in the absence of further knowledge one could implement up to 20 population splits, and no base chance, which may improve the likelihood of a successful outcome.

In the future, selection strategies could be further refined with the aid of novel landscape inference methods, which can predict landscape parameters from sequence alignment [54] or experimental measurements of fitness in real-time as they are collected [39, 55]. Further work will be required on translating landscape inference to optimal strategy, which may either consider how they relate to NK landscapes (or other parameterisations of landscape ruggedness), or by conducting experiments to compare landscape inference outputs and directed evolution strategy performance. To summarise our recommendation, we propose two possibilities for the future (Fig 5E). With no prior data about the landscape available, the default NK strategy outlined above could be implemented. In order to refine the process further, landscape inference tools could be used to generate information on the ruggedness of their landscape and select an appropriate strategy.

## Discussion

This study demonstrates that even in the absence of sequencing information, there are approaches that can be used to improve directed evolution outcomes. Such approaches were demonstrated both on simulated NK landscapes, and on protein empirical fitness landscapes. On the GB1 landscape, we showed alternative directed evolution strategies can lead to up to a 19-fold increase in the probability of reaching the global optimum, without any requirement for sequencing data.

There are several ways in which the methods outlined in this paper could be implemented experimentally. In its simplest form, a selection function of just "fitness threshold" and "base chance" parameters (Fig 2A) could be implemented by pipetting the appropriate portion of pre-selection cells into a post-selection population. A more sophisticated approach could incorporate the selection function into the Fluoresence Activated Cell Sorting (FACS) methodologies, or microfluidic-based methods for cell selection and sorting [16, 17]. Population splitting is the most intuitive to implement, and in many cases is already performed, in the form of biological replicates. Multi-dimensional selection in the form we describe can only be implemented with sorting technology that allows for multiple properties to measured prior to selection (i.e. with microfluidics) [16, 17].

The main limitation of this work is that optimal strategies are dependent on the shape of the fitness landscape, which is generally unknown. With the continued development of fitness landscape inference tools, however, it is becoming possible to make estimations about landscape properties. Of these there are two forms of fitness landscape inference we wish to highlight: Those that employ pre-existing sequence data, and those that are based on data as it is collected *during* a directed evolution experiment. The former utilises multiple sequence alignment (MSA) information, and identifies interacting residues from patterns of co-evolution (the frequency of interacting residues being an indicator of ruggedness). This approach is employed by EVmutation [51] and DeepSequence [52] to predict the effects of novel

mutations. It is also employed in SLIP [54], which uses MSA information to generate synthetic landscapes. These methods are not currently accurate enough in isolation to predict sequence-to-function fitness peaks reliably, but if used in combination with directed evolution (i.e. to inform ruggedness-informed selections of base chance and splitting, as well as promising starting points), they could produce a superior approach overall. The latter form of fitness landscape inference methods are those that take place during a directed evolution experiment and hence require no such bank of sequencing data to produce an MSA. Examples of such tools utilise both intermittent sequencing [39], and phenotypic measurements alone [55] to build a picture of the landscape as evolution takes place. The phenotypic approach uses a trained neural network to make ruggedness predictions from the behaviour of a randomly mutating population. Such an approach would be particularly powerful in combination with the methods outlined in our study, as the entire pipeline could be conducted with no sequencing required (neither prior MSA data nor sequencing from during the experiment). Work remains to be done, however, on bridging the gap between the output of each landscape inference method and selection of the most promising directed evolution strategy.

Another limitation of this study is the reliance on NK models. We chose to use such parameterisable fitness landscapes for testing because empirical landscapes, such as that of GB1 and TrpB, are scarce. NK models do not perfectly represent the statistical properties of natural fitness landscapes, therefore in order to improve the reliability of simulations such as these, future work could tailor the NK model to the specific application of protein fitness landscapes. For instance, by allowing K to be drawn from a distribution for each amino acid position, as opposed to a constant value, or by integrating the information offered by PAM substitution matrices into fitness estimations. NK model variants also exist that emphasise the role of neutral drift, another factor that could be integrated into an alternative NK model [49, 50].

With *de novo* protein design approaches still in their infancy, directed evolution remains an important component of the protein engineering toolbox. Here we demonstrate that the standard approach to directed evolution, which is to select the top variants from a single population, can be sub-optimal both in real and synthetic fitness landscape contexts. We propose two strategies to overcome the limitations of the standard approach; using base chance and population splitting to increase landscape exploration. By combining these techniques we observe that up to a 19-fold increase in the probability of reaching the global optimum is made possible. As our knowledge of how to make predictions about directed evolution strategy develops, most notably with the integration of landscape inference tools, it may become possible to unlock further such improvements in directed evolution without the need for sequencing.

## Methods

### Model

To test our selection algorithms we implemented *in silico* simulations of directed evolution, which can be applied to either synthetic or empirical landscapes. We model each genetic variant within a population using a one-dimensional array of $N$ integer values (Fig 1), which is initialised at a random starting location (NK) or wildtype sequence (GB1, TrpB), creating a population of size $P$. To calculate the corresponding fitness value of a gene, this array is either used as input to the NK model (Methods: NK Landscapes), or used as coordinates to look up fitness in an empirical data set (Methods: Empirical Landscapes). Our simulation algorithm proceeds with three cyclic steps; selection, proliferation, and mutation.

Individual cell fitness values are input to a selection function (Fig 1), which translates relative fitness into probability of selection. Then, each cell is selected (or not) based only on its respective probability of selection. Cells that are selected are proliferated to bring the

population back up to its original size. To perform proliferation, a *new* population of size $P$ is created by randomly sampling (with replacement) from the previously selected cells. This introduces a degree of stochasticity mirroring experimental error and biological variation, as described in other models of evolutionary processes [56].

Once a population of size $P$ has been selected and proliferated, mutations are introduced by making random changes to genes in the population. For every cell's genetic code (an array), each residue (i.e. nucleotide or amino acid) has a random chance, $p_I$, of changing into another random *different* value (i.e. each other possibility is equally likely). Given the gene length $N$, the expected number of changes across the entire gene (the "mutations per cell") is given by $\mu = Np_I$. This quantity is useful to work with, as a fixed $\mu$ will give comparable results as $N$ varies.

The final result is a new population of size $P$, and the process of selection, proliferation, and mutation may be repeated again. In this work, it is assumed that this runs for a fixed number of iterations, and the final result (by which we compare methods) is the maximum fitness across the final population. "Normalised fitness" divides this measure by the global maximum of the landscape.

## Selection functions

In this work, we introduce the concept of a "selection function". This function takes a cell's fitness percentile, defined as the percentage of other cells in the population it is fitter than, and outputs its chance of being selected to be in the next population. The selection function used in this work is defined by two parameters: a "fitness threshold" and a "base chance" (Fig 2A). Above the "fitness threshold" the selection function outputs 1, otherwise it outputs the "base chance" (Eq 1). In this work, selection functions are normalised to select a constant fraction of cells (20% throughout this paper). This is to ensure that in a real continuous directed evolution experiment, the time required for proliferating cells between iterations remains effectively constant such that a "fair" comparison is made between strategies in terms of performance per-unit-time. By fixing the selected proportion (given by the integral of the selection function), our parameter search space is reduced to one dimension, as every base chance has only one possible corresponding fitness threshold value (Eq 2).

$$selection\ chance = \begin{cases} base\ chance & \text{if } fitness\ percentile < fitness\ threshold \\ 1 & \text{if } fitness\ percentile \geq fitness\ threshold \end{cases} \tag{1}$$

$$fitness\ threshold = \frac{1 - selected\ fraction}{1 - base\ chance} \tag{2}$$

## NK landscapes

The NK model is a widely used approach for generating synthetic fitness landscapes with tuneable ruggedness [42, 43]. In the NK model, a gene is represented by an array of $N$ sites, each of which has $A$ possible values. In this work we will generally set $A = 2$ such that each entry has two possibilities (i.e. binary 1 or 0). Every entry corresponds to a "locus", which interacts with $K$ other loci in the gene. The fitness contribution of each locus is dependent on the state of that locus and the state of the $K$ other loci it interacts with. The fitness $F$ of a gene $G$ is the sum of the fitnesses of each locus. When $K = 0$, all loci are independent and the model is linear (and hence has a single peak). The other extreme, $K = N − 1$, in which every locus interacts with

every other locus, is maximally unstructured; the fitness landscape consisting of only random noise.

The NK model defines a parameterised *distribution* over functions $F : A^N \rightarrow \mathbb{R}$. To define how to generate samples from this distribution, first let $L : N \times K \rightarrow N$ be a "locus function", where $L(a, i)$ is the $i$th site that locus $a$ interacts with. We require each locus to interact with precisely $K$ other, uniformly random, sites, independent of all other loci. We then also have $N \cdot A^{K+1}$ independent, identically distributed standard normal ($\mathcal{N}(0, 1)$) random variables, which we denote as $X^a_{i_0, i_1, i_2 \dots i_K}$, where $a \in \{1 \dots N\}$ and $i_{0,1, \dots, K} \in \{1 \dots A\}$. These represent the possible fitness contributions of each loci, with $a$ being the index of the loci in question, and $i_{0,1,\dots,K}$ being the values of that loci and the $K$ other loci it interacts with, plus itself ($i_0$).

Then, $F$ is given as the following, where $g \in A^N$, and $g[i]$ denotes the $i$th site in $g$:

$$F(g) = \sum_{a=1}^{N} X^a_{g[a], g[L(a,1)], g[L(a,2)], \dots, g[L(a,K)]} \tag{3}$$

Computationally, the algorithm used explicitly generates and stores $L$; in particular as a $N \times N$ binary matrix. However, it does not store the $N \cdot A^{K+1}$ random variables explicitly as doing so would require large amounts of memory. Instead, every time the value of $X^a_{i_0, i_1, i_2 \dots i_K}$ is required, $a, i_0, i_1, \dots, i_K$ are used as inputs to a pre-defined deterministic pseudorandom generation algorithm.

## Empirical landscapes

Empirical fitness landscapes are real data sets produced by measuring the fitness of all possible sequential variants of a protein (or region of a protein). The empirical landscapes used in this paper are GB1 [23] and TrpB [25]. For our algorithm, the landscapes (each four amino acid positions) are stored as four-dimensional arrays, where each dimension corresponds to a variable residue. By assigning every amino acid a number from 1 to 20, each sequence of amino acids can therefore be mapped to a set of coordinates that points to the corresponding fitness value in the array. All directed evolution simulations were started from the wild-type sequence "VDGV" for GB1, and "VFVS" for TrpB.

## Supporting information

**S1 Fig. Competition plots showing base chance performance over varying NK landscapes.** Data shows the proportion of runs that are won by either a 0 base chance, or 0.1 base chance strategy over different landscapes. Where the outcome is tied, the value is 0. Value of N increases vertically, and the value of K/N increases up horizontally. Mutations per cell = 0.1, population size = 1000.
(TIF)

**S2 Fig. Competition plots showing the influence of noise on base chance performance over 100 iterations.** To introduce noise, values sampled from $\mathcal{N}(0, noise^2)$ are added to fitness values each generation. N = 25, K = 5, mutations per cell = 0.1, population size = 1000.
(TIF)

**S3 Fig. Comparing the performance of population splitting across landscapes of increasing ruggedness.** Mutations per cell = 0.1.
(TIF)

**S4 Fig. Parameter sweep over population sizes (5000,1000,100), mutation rates (0.05/N, 0.1/N, 0.15/N, 0.2/N, 0.25/N), N values (50, 25) and K values (0.25N, 0.2N, 0.15N, 0.1N,**

**0.05N, 0).** Results were processed by ranking each of the 30 strategies (combinations of base chance and splitting) against one another for each set of parameters. A: Average rank overall. B: Histogram of ranks, top performing strategy vs standard strategy (no base chance, no splits).
(TIF)

## Author Contributions

**Conceptualization:** Jessica James, Sebastian Towers, Jakob Foerster, Harrison Steel.

**Data curation:** Jessica James, Sebastian Towers.

**Formal analysis:** Jessica James, Sebastian Towers.

**Funding acquisition:** Harrison Steel.

**Investigation:** Jessica James, Sebastian Towers.

**Methodology:** Jessica James, Sebastian Towers.

**Project administration:** Jessica James, Sebastian Towers, Jakob Foerster, Harrison Steel.

**Resources:** Jakob Foerster, Harrison Steel.

**Software:** Jessica James, Sebastian Towers.

**Supervision:** Jakob Foerster, Harrison Steel.

**Validation:** Jessica James, Sebastian Towers.

**Visualization:** Jessica James, Sebastian Towers.

**Writing – original draft:** Jessica James, Sebastian Towers.

**Writing – review & editing:** Jessica James, Sebastian Towers, Jakob Foerster, Harrison Steel.

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
