## [Decision Letter · Decision Letter 0]

21 May 2024

Dear Mr Steel,

Thank you very much for submitting your manuscript "Optimisation strategies for directed evolution without sequencing" for consideration at PLOS Computational Biology.

As with all papers reviewed by the journal, your manuscript was reviewed by members of the editorial board and by several independent reviewers. In light of the reviews (below this email), we would like to invite the resubmission of a significantly-revised version that takes into account the reviewers' comments.

Please address all the major comments of reviewer 2 carefully. Without satisfactory replies, the paper may be judged to be more suitable for a more specialized journal.

We cannot make any decision about publication until we have seen the revised manuscript and your response to the reviewers' comments. Your revised manuscript is also likely to be sent to reviewers for further evaluation.

Sincerely,

Alexandre V. Morozov, Ph.D.

Academic Editor

PLOS Computational Biology

Zhaolei Zhang

Section Editor

PLOS Computational Biology

Reviewer's Responses to Questions

**Comments to the Authors:**

Reviewer #1: In this paper, the authors present several approaches for improving the efficiency and performance of directed evolution experiments, without a need for detailed sequencing data. This is achieved through the implementation of a threshold for selection and a “base chance” parameter that enables exploration and exploitation to be tuned, and the inclusion of population splitting to enable the exploration of multiple different paths that a population can take through the fitness landscape. Using theoretical and empirical fitness landscapes they demonstrate that their approach outperforms standard approaches.

Overall, I found this to be an enjoyable and mostly well-written paper that tackles an increasingly important challenge in bioengineering. The work is perfectly aligned to the readership of PLOS Computational Biology, and I believe will be an important contribution to the field. Specifically, the ability for these relatively simple additions to greatly improve the chance of finding optimal solution during a directed evolution experiment make this work of great value and ensure its broad appeal to both theoretical and experimental audiences. I found the presentation of the results to be very clear and had no issue following the models and selection algorithms presented. I did have some more substantial comments (see below) on the scale of the theoretical simulations and access to code, but I believe these would require only a minor revision to address.

My main comments were:

– It was unclear to me why N = 25 and K = 5 was used throughout to assess the selection scheme. It seems likely that features like population splitting, will play an important role in landscapes that are not too rugged or smooth. Better understanding that transition would be interesting to explore.

– I found the population splitting aspect of the work very interesting and wondered whether there was a simple mathematical relationship that could be used to calculate the optimal number of sub-populations from the population size, N and K values. The data in Fig 3D looks smooth and so there might be quite a simple form that could be established. This would be helpful in making good choices, especially as N and K can be estimated from experimental data as the processes of evolution takes place.

– I could not work out how to run the models in the linked code repository. It may be helpful to have a notebook that covers all the key types of simulation performed in the work to enable reproduction of the results and use by others.

I also had several minor comments that need to be considered:

L14: “biased toward user-defined desirable variants” -> “toward desirable user-defined traits”

L19: “desirable output values” -> “desirable output expression levels”

L24: Perhaps introduce the term “screening” that is typically used in the context discussed?

L28: “increasingly high-dimensional information” – I agree that the data has a higher dimension (i.e., includes time), but not that the dimension is increasing beyond that. Clarifying this would be helpful.

L36: Does the trait-of-interest undergo directed evolution? Is it not the biological entity?

L43: “variants with each iteration” -> “variants during each iteration”

L69: “which in particular enable effective” -> “which enable effective”

L84: “e.g. FACS” -> “e.g. via FACS”

L119: “of the distribution Fig. 2C” -> “of the distribution. Fig. 2C”

L160: “dataset to to reduce” -> “dataset to reduce”

– It may be useful to have a short paragraph in the introduction giving some background on emerging in vivo mutagenesis methods.

Figure 2: For the normalised fitness measures, it would be helpful to see the spread that is associated with the average presented. I assume it would be feasible to include the standard deviation in these values? I’m wondering how this is affected by the parameters.

All Figures: While the content of the figures is excellent and well-presented, the text labels in most of them was often too small to read comfortably. I would suggest considering increasing the size of the text in all of them to ensure it is 8pt when present in the paper format and possibly carrying out minor reorganisation/resizing of the panels where this is challenging.

Overall, a very interesting computational study that I believe will be of great value to the directed evolution and bioengineering communities at large.

Reviewer #2: In this work, James and collaborators present a thoughtful discussion of strategies for optimizing directed evolution experiments that do not rely on sequencing. Using a variety of simulations, they show that tweaking different features of the directed evolution process, including the mutation rate, method for selection, and number of populations, can be advantageous in different scenarios. This is a good point, which experimentalists should take to heart.

While the concept presented here is interesting, it’s a bit more difficult to connect that concept with practice. The authors present some work based on real data, but I have questions about this process as well. My major comments are below.

1. Application in practice. The authors describe a number of different parameters that can be tuned to affect directed evolution experiments, including varying mutation rates, selection functions, and so forth. However, which choices are “better” and which ones are “worse” depend on the underlying fitness landscape. For example, when using a step function for selection, increasing the base chance of selection is helpful for rugged landscapes but harmful for smooth ones. In Figure 3, the outcome of the simulated experiments is a nonlinear function of the number of subpopulations, with the optimal performance found somewhere between the maximum and minimum numbers. Beyond knowing that these variables are ones that could, in principle, be experimentally manipulated, how would the results described here be used in practice?

2. Shape of the fitness landscape. Several different landscapes are used in this work, including NK landscapes with different parameters and an empirical landscape derived from ref. 19. The best approach to optimize directed evolution depends upon the shape of the landscape. Are there typical landscapes that experimentalists are most likely to encounter? If so, what evidence supports this? How would one know the shape of the landscape, and thus what directed evolution approach to use, in a novel experiment?

3. Fitness in simulations. It appears that the authors assume that some metric of fitness can be measured without noise in realistic data. This seems unlikely. How would these methods work if precise fitness measurements were not available? For example, one could consider some measurable quantity (e.g., fluorescence intensity) to be a noisy measurement of a true, underlying component of fitness.

4. Application to real data. While the simulation using an empirical landscape suggests that alternative experimental approaches could improve the results of directed evolution, this is far from a proof in practice. When the shape of the fitness landscape is known, it is not surprising that it is possible to generate post hoc an experimental process that more reliably approaches the fitness maximum. For a novel data set this would be challenging. As discussed above, different experimental choices can either help or impede evolution, depending on the (unknown) shape of the underlying landscape. Without an application, it is difficult to appreciate the utility of this approach.

Minor points:

1. Figure 2 C and D are reversed in the caption.

2. The term in vivo mutagenesis is a bit confusing. After reading the linked review, I understand: this refers to mutation within a replicating cell, rather than artificial mutagenesis to generate phenotypic variation. But the directed evolution experiment itself is certainly in vitro. It may be helpful to add a sentence that explains this concept more precisely.

**Have the authors made all data and (if applicable) computational code underlying the findings in their manuscript fully available?**

Reviewer #1: **No: **There is a code repository, but it needs some better documentation to understand how it was used for this study.

Reviewer #2: Yes

PLOS authors have the option to publish the peer review history of their article (what does this mean?). If published, this will include your full peer review and any attached files.

Reviewer #1: No

Reviewer #2: No
---

## [Decision Letter · Decision Letter 1]

26 Aug 2024

Dear Prof Steel,

Thank you very much for submitting your manuscript "Optimisation strategies for directed evolution without sequencing" for consideration at PLOS Computational Biology.

As with all papers reviewed by the journal, your manuscript was reviewed by members of the editorial board and by several independent reviewers. In light of the reviews (below this email), we would like to invite the resubmission of a significantly-revised version that takes into account the reviewers' comments.

Please address the concerns of reviewer 2 carefully, otherwise the paper may be found unsuitable for PLOS Comp Bio.

We cannot make any decision about publication until we have seen the revised manuscript and your response to the reviewers' comments. Your revised manuscript is also likely to be sent to reviewers for further evaluation.

Sincerely,

Alexandre V. Morozov, Ph.D.

Academic Editor

PLOS Computational Biology

Zhaolei Zhang

Section Editor

PLOS Computational Biology

Reviewer's Responses to Questions

**Comments to the Authors:**

Reviewer #1: I commend the authors on fully considering and addressing my concerns and strengthening this work. I am happy to now support for publication.

Reviewer #2: While the authors have responded to prior comments, significant doubts remain. The most essential point is how the methods the authors recommend could be applied in practice when the shape of the fitness landscape is unknown.

There are several comments made in the response to reviewers which did not appear to make it into the paper, but in the Discussion the authors state that “the continued development of fitness landscape inference tools, however, we may be able to estimate such parameters in advance (or during) a directed evolution experiment.” At the level of discussion in the paper, this seems circular: if the landscape is already known by other means, then what is the purpose of the experiment? Can this point be clarified?

More generally, the authors should precisely articulate how their ideas should be used, as well as the evidence for them. If, as stated in the response, there are some approaches that should be useful in general (regardless of the shape of the landscape), then the authors should say this clearly and support this claim with evidence. Of course, this does not imply that methods must be tested against all possible landscapes, which might as well be infinite in number. Simulations using a variety of easily parameterized landscapes would be sufficient to argue for the general use of a method.

If, on the other hand, the authors do not believe (or do not have evidence to support) the general applicability of different methods, then this should also be discussed clearly in the paper. Alternately, if the idea is that experimentalists should learn something about the landscape before implementing some of the approaches described here – which seems to be the argument presented in the Discussion in the current version of the paper – then the authors should elaborate on this point more explicitly. How would they envision practitioners applying the population splitting/base chance ideas in a new experiment?

**Have the authors made all data and (if applicable) computational code underlying the findings in their manuscript fully available?**

Reviewer #1: Yes

Reviewer #2: Yes

PLOS authors have the option to publish the peer review history of their article (what does this mean?). If published, this will include your full peer review and any attached files.

Reviewer #1: No

Reviewer #2: No
---

## [Decision Letter · Decision Letter 2]

4 Dec 2024

Dear Prof Steel,

We are pleased to inform you that your manuscript 'Optimisation strategies for directed evolution without sequencing' has been provisionally accepted for publication in PLOS Computational Biology.

Best regards,

Alexandre V. Morozov, Ph.D.

Academic Editor

PLOS Computational Biology

Zhaolei Zhang

Section Editor

PLOS Computational Biology

Feilim Mac Gabhann

Editor-in-Chief

PLOS Computational Biology

Jason Papin

Editor-in-Chief

PLOS Computational Biology

Reviewer's Responses to Questions

**Comments to the Authors:**

Reviewer #2: I thank the authors for their thorough revisions, which have clearly addressed concerns about how their ideas could be implemented in practice (and how this information is communicated to the audience). I am happy to support publication of the manuscript at this stage.

**Have the authors made all data and (if applicable) computational code underlying the findings in their manuscript fully available?**

Reviewer #2: Yes

PLOS authors have the option to publish the peer review history of their article (what does this mean?). If published, this will include your full peer review and any attached files.

Reviewer #2: No

---

## [Editor Report · Acceptance letter]

12 Dec 2024

PCOMPBIOL-D-24-00462R2 

Optimisation strategies for directed evolution without sequencing

Dear Dr Steel,

I am pleased to inform you that your manuscript has been formally accepted for publication in PLOS Computational Biology. Your manuscript is now with our production department and you will be notified of the publication date in due course.

With kind regards,

Zsofia Freund
